# Loss of ZBTB20 impairs circadian output and leads to unimodal behavioral rhythms

Zhipeng Qu[1†], Hai Zhang[2†], Moli Huang[3], Guangsen Shi[1], Zhiwei Liu[3], Pancheng Xie[1], Hui Li[2], Wei Wang[3], Guoqiang Xu[4], Yang Zhang[3], Ling Yang[3], Guocun Huang[5], Joseph S Takahashi[5], Weiping J Zhang[2*], Ying Xu[1,3*]

[1]MOE Key Laboratory of Model Animal for Disease Study, Model Animal Research Center, Nanjing University, Nanjing, China; [2]Department of Pathophysiology, Second Military Medical University, Shanghai, China; [3]Cambridge-Suda Genomic Research Center, Soochow University, Suzhou, China; [4]College of Pharmaceutical Sciences, Soochow University, Suzhou, China; [5]Department of Neuroscience, University of Texas Southwestern Medical Center, Dallas, United States

**Abstract** Many animals display morning and evening bimodal activities in the day/night cycle. However, little is known regarding the potential components involved in the regulation of bimodal behavioral rhythms in mammals. Here, we identified that the zinc finger protein gene *Zbtb20* plays a crucial role in the regulation of bimodal activities in mice. Depletion of *Zbtb20* in nerve system resulted in the loss of early evening activity, but the increase of morning activity. We found that *Zbtb20*-deficient mice exhibited a pronounced decrease in the expression of *Prokr2* and resembled phenotypes of *Prok2* and *Prokr2*-knockout mice. Injection of adeno-associated virus-double-floxed *Prokr2* in suprachiasmatic nucleus could partly restore evening activity in *Nestin-Cre; Zbtb20*[fl/fl] (NS-ZB20KO) mice. Furthermore, loss of *Zbtb20* in *Foxg1* loci, but intact in the suprachiasmatic nucleus, was not responsible for the unimodal activity of NS-ZB20KO mice. Our study provides evidence that ZBTB20-mediated PROKR2 signaling is critical for the evening behavioral rhythms.

*For correspondence: wzhang@ smmu.edu.cn (WJZ); yingxu@suda. edu.cn (YX)

[†]These authors contributed equally to this work

## Introduction

The circadian clock orchestrates the daily rhythms in physiology and behaviors that allow organisms to anticipate regular environmental cycles and increase their adaptive fitness. The rhythms originate from a central circadian pacemaker located in the suprachiasmatic nucleus (SCN) of the hypothalamus, and are synchronized to the environmental light-dark cycle via the retinohypothalamic tract (*Morin and Allen, 2006*; *Reppert and Weaver, 2002*; *Vansteensel et al., 2008*). At the molecular level, the rhythms are generated in a cell-autonomous manner by transcription–translational-based feedback loops, which are composed of clock proteins, such as PERIOD (PER1, PER2, and PER3), CRYPTOCHROME (CRY1 and CRY2), CLOCK, BMAL1, NR1D1, NR1D2, and ROR (RORα, RORb, and RORg) in mammals (*Mohawk et al., 2012*). At the cellular level, the SCN is a heterogeneous structure consisting of multiple types of neurons (*Antle and Silver, 2005*), which secretes more than 100 identified neurotransmitters, neuropeptides, cytokines, and growth factors (*Abrahamson and Moore, 2001*; *Lee et al., 2010*). Their coherence, in turn, imparts overt rhythms of activity-rest, physiology, and metabolism across the entire organism (*An et al., 2013*; *Herzog, 2007*; *Maywood et al., 2006*).

The SCN is absolutely required for behavioral rhythmicity, with a bimodal activity pattern consisting of morning and evening activity (*De la Iglesia et al., 2000*; *Grima et al., 2004*; *Helfrich-Förster, 2009*; *Inagaki et al., 2007*; *Jagota et al., 2000*; *Kon et al., 2014*; *Stoleru et al., 2004*), driving activity onset to adapt to dusk and driving the end of activity at dawn in nocturnal animals

(*Aschoff, 1966*; *Inagaki et al., 2007*; *Jagota et al., 2000*; *Kon et al., 2014*). Prokineticin-2 (*Prok2*) is a clock-controlled gene, and PROK2 signaling via prokineticin receptor-2 (PROKR2) is a suprachiasmatic nuclei clock output signal that has been implicated in the redistribution of activity from early to late circadian night (*Cheng et al., 2002*; *Li et al., 2006*; *Prosser et al., 2007*). However, little is known regarding its regulation, which corresponds to evening and morning behavioral activities in mammals. Therefore, transcription factors that regulate the expression of *Prok2* or *Prokr2* signaling are necessary to expand the understanding of bimodal activity for entrainment and adaptation to seasonal changes in the environment.

ZBTB20 is a member of the bric-a-brac tramtrack broad complex/poxvirus and zinc domain family, which functions primarily as transcriptional factors via interactions mediated by their conserved C2H2 Krüppel-type zinc finger and BTB/POZ domains (*Mitchelmore et al., 2002*; *Zhang et al., 2001*). Previous studies have indicated that ZBTB20 could be involved in metabolism, development, growth, glucose homeostasis, and immune responses (*Liu et al., 2013*; *Ren et al., 2014*; *Sutherland et al., 2009*; *Xie et al., 2010*, *2008*; *Zhang et al., 2015*, *2012*). More importantly, missense mutations of ZBTB20 have been linked to Primrose syndrome (*Cordeddu et al., 2014*), suggesting that transcription factor ZBTB20 is an essential element for neurological disorders. Here, we found that mice lacking *Zbtb20* exhibited impaired evening activity rhythms both in 12-hr light/12-hr dark (LD) cycles and under constant darkness conditions (DD). There are a limited number of functional genes that can be meaningfully correlated with evening activity or morning activity in mammals. To our knowledge, *Zbtb20*-knockout mice are the only model that mimics the phenotype of *Prok2* and *Prokr2*-knockout mice (*Li et al., 2006*; *Prosser et al., 2007*). Interestingly, both *Prokr2* transcript level and protein level were significantly reduced in *Zbtb20*-depleted mice. Moreover, ZBTB20 directly regulated the transcription of *Prokr2*. The overexpression of PROKR2 in *Cre*-mediated *Zbtb20*-knockout mice partly restored evening activity. These data strongly suggested that ZBTB20 and PROKR2 are key pathway for the evening behavior. Finally, we unexpectedly found that ZBTB20 globally regulates mitochondrial, transportation and neurodegenerative enrichment pathways.

## Results

### Loss of ZBTB20 impairs evening activity but increases morning activity

In a systematic phenotyping project, we generated a conditional *Zbtb20* allele using the *Cre-loxp* recombination system (*Figure 1A*). Mice carrying *Zbtb20*^fl/fl were mated with those with the *Nestin-Cre* transgene to generate *Zbtb20*-deficient mice (hereafter called NS-ZB20KO). *Nestin-Cre* is well documented in both neural stem cells and radial glia (*Tronche et al., 1999*). In the NS-ZB20KO mice, the amount of *Zbtb20* estimated by quantitative RT-PCR (Q-PCR) was reduced by 90% in the SCN, 70% in the olfactory bulb and 90% in the cerebellum in NS-ZB20KO mice, with no change observed in *Zbtb20* expression in the liver (*Figure 1—figure supplement 1*). Based on immunofluorescence staining, ZBTB20 protein was abundantly expressed in the SCN neurons from WT mice, but was almost undetectable in NS-ZB20KO mice determined by immunofluorescence staining (*Figure 1B*). Western blot analysis using anti-ZBTB20 antibodies revealed that expression of ZBTB20 was markedly reduced but not completely abolished in the hypothalami of NS-ZB20KO mice, potentially due to non-*Nestin*-expressing cells (*Figure 1—figure supplement 1*). NS-ZB20KO mice were normal at birth, but they began to display a range of abnormalities, including decreased body weight, body trembling, and hyperactivity after midnight (*Video 1* and *Video 2*). A significant proportion of NS-ZB20KO mice died at approximately 3–4 months. Except for reduced brain size and weight, autopsy failed to reveal obvious alterations except for reduced brain size and weight in NS-ZB20KO mice compared to those in *Nestin-Cre* transgenic mice or wild-type littermates.

Having found that NS-ZB20KO mice displayed abnormal behavior, we monitored the wheel-running activity of NS-ZB20KO mice, along with controls including *Zbtb20*^fl/fl mice, which lacked the *Nestin-Cre* transgene, and *Nestin-Cre; Zbtb20*^fl/+ mice, which had one intact *Zbtb20* allele and *Nestin-Cre*. Under LD cycles, both types of control mice exhibited normal bouts of restricted activity during the dark phase (*Figure 1C*). Under DD conditions, the control mice exhibited normal free-running periods (*Figure 1C*). However, NS-ZB20KO mice displayed a dramatic loss of the early evening activity under both LD and DD conditions, along with significantly limited durations in the early

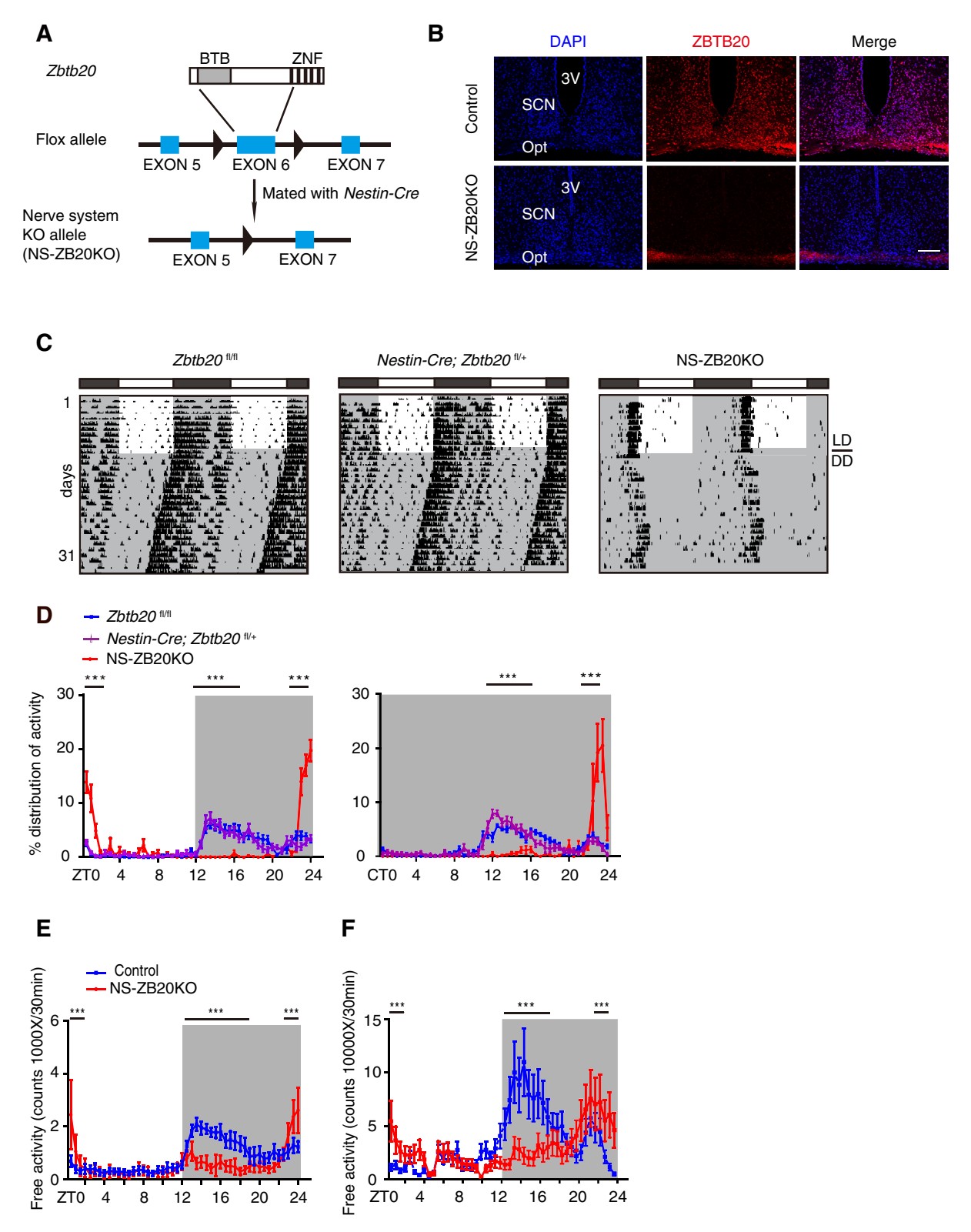

**Figure 1.** Loss of *Zbtb20* alters evening activity and morning activity. (**A**) Targeting strategies for *Zbtb20*. Exon 6 of the *Zbtb20* gene was flanked by two loxP sites as indicated, and it was deleted by CRE recombination in the nervous system. (See also *Figure 1—figure supplement 1*). (**B**) Immunofluorescence staining for ZBTB20 in SCN from control and NS-ZB20KO mice (slices from at least three mice per group). (Scale bar, 100 μm). (**C**) Representative actograms of wheel-running activities from control (n = 15), *Nestin-Cre; Zbtb20*fl/+ (n = 6) and NS-ZB20KO mice (n = 13). The mice were

*Figure 1 continued on next page*

*Figure 1 continued*

first entrained to an LD cycle for 10 days and then released in DD for approximately 3 weeks. Black shading indicates the time when lights were off, and the white box indicates the time when lights were on. (**D**) Percentage distributions of wheel-running activity in LD or DD from control (blue, n = 15), *Nestin-Cre; Zbtb20*fl/+ (purple, n = 6) and NS-ZB20KO (red, n = 13) mice derived from (**C**). Values for wheel-running activity were accumulated for data on days 3–10 in LD and on days 11–18 in DD, respectively. The results are plotted as the mean ± SEM, ***p<0.001 by one-way ANOVA with multiple comparisons and Tukey's test. (**E,F**) Temporal profiles of spontaneous activity recorded by CLAMS (**E**) and miniature telemetry (**F**) in LD. The spontaneous activity from 3 consecutive days of the LD cycle was accumulated in bins of 20 min and averaged to produce one 24 hr profile. The number of mice for CLAMS: control n = 12; NS-ZB20KO n = 9; for miniature telemetry: control n = 9; NS-ZB20KO n = 5. Each point represents the mean ± SEM, ***p<0.001 by one-way ANOVA with multiple comparisons and Tukey's test (also see *Figure 1—figure supplement 1*).

The following source data and figure supplement are available for figure 1:

**Source data 1.** Data for temporal profiles of wheel-running activity in control and NS-ZB20KO mice.
**Source data 2.** Data for temporal profiles of spontaneous activity in control and NS-ZB20KO mice.
**Figure supplement 1.** Deletion of *Zbtb20* in brains from NS-ZB20KO mice.

morning activity (*Figure 1C and D*, red peak). Free-running periods of the activity offset were 23.76 ± 0.15, n = 15 for control mice and 24.07 ± 0.09, n = 13 for NS-ZB20KO mice (p=5.09175E-05).

To confirm the phenotype of abnormal activity, the spontaneous locomotor activity was monitored using the Comprehensive Lab Animal Monitoring System (CLAMS) and miniature telemetry. Consistent with the wheel-running activity, the control mice displayed conspicuous evening activity and morning activity, whereas NS-ZB20KO mice showed obviously reduced evening activity (*Figures 1E and F*) under the LD cycles (CLAMS: n = 8, p=0.0078; DSI: n = 9, p=5.1E-07), while the activity counts were significantly increased in NS-ZB20KO mice compared with control mice during the morning phase (*Figures 1E and F*). None of these effects on locomotor activity could be attributed to general deficits in activity ability because NS-ZB20KO mice showed hyperactivity during the morning phase (*Figures 1C–F*, *Video 2*). These phenotype analyses revealed that ZBTB20 is essential for the sustained generation of evening activity rhythms.

## The defects in light responsiveness in NS-ZB20KO mice

To evaluate photic resetting and entrainment in NS-ZB20KO mice, we observed the wheel-running behavior of control and NS-ZB20KO mice subjected to different light/dark schedules. We found that the phase-delaying effects caused by applying saturating light pulses at CT12-CT18 were significantly smaller in NS-ZB20KO mice than in control mice (WT: −3.18 ± 0.48 hr n = 6; NS-ZB20KO; −2.31 ± 0.36 hr n = 10, respectively) (*Figure 2A*). In contrast, there were no significant differences between genotypes in the phase advances induced by saturating light pulses at CT20-24 (*Figure 2B*). However, in response to a 6 hr phase advance of the LD cycle, the activity offset in NS-ZB20KO mice readjusted to the new LD regimen much more slowly than that in the control mice (*Figure 2C*). The activity end of the control mice seemed to be masked by a light phase of phase advanced LD cycle in transients, which might be taken as a rapid phase-shift. On the other hand, the morning component was not masked probably because of enhanced morning activity in NS-ZB20KO mice (*Figure 2C*). In response to 6 hr phase-delay of the LD cycle, the control mice showed a gradual delayed offset of activity as expected (*Figure 2D*), while NS-ZB20KO mice ended their activity significantly earlier on day 1,

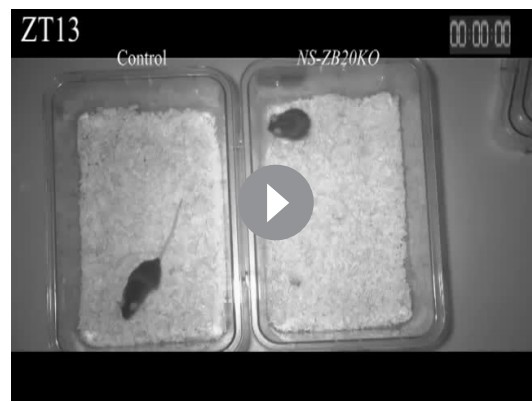

**Video 1.** Activity of control and NS-ZB20KO mice at ZT13 (early night).

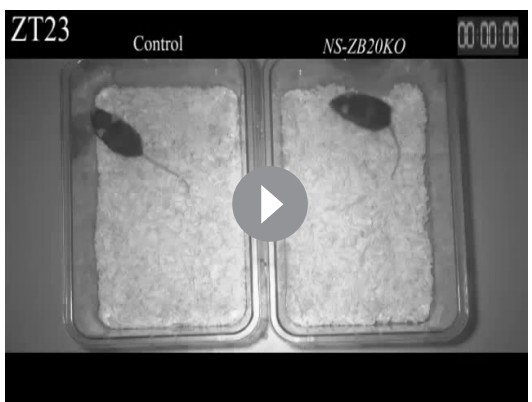

**Video 2.** Activity of control and NS-ZB20KO mice at ZT23 (early morning).

suggesting a defect in the immediate adaptation to a new LD cycles in NS-ZB20KO mice. There were no significant difference in the phase delay for the subsequent days between genotypes (*Figure 2D*). Indeed, the offset of activity showed a widened variation compared with the onset of activity between individual mice. We are uncertain whether lack of significant difference in the re-entrainment to a phase-delay shift of LD cycles in the subsequent days might result from counteraction by variation.

Circadian oscillators drive activity onset and offset in response to dawn and dusk, respectively (*Enoki et al., 2012*; *Inagaki et al., 2007*; *Yamaguchi et al., 2003*). The loss of the early evening activity and the impairment of light-pulse-induced phase shifting prompted us to ask whether these mice could adjust to a long day (LD 18:6) or a short day (LD 6:18). We found that both control and NS-ZB20KO mice could be clearly entrained to a long day (*Figure 2E and G*). The compressed activity bands were also observed as previously described for a long day (*Inagaki et al., 2007*). Interestingly, NS-ZB20KO mice had a difficulty in entraining to a short day (*Figure 2F*). The time interval between the activity offset and light off was significantly different between genotypes for LD 6:18 (p<0.01) (*Figure 2G*). Thus, loss of *Zbtb20* shapes the behavioral response to light perturbation.

## Loss of *Zbtb20* affects circadian output pathway

As NS-ZB20KO mice displayed circadian behavioral defects and entrainment impairment, we wondered whether the loss of *Zbtb20* affects the core circadian oscillator or the pathway that translates signals from the clock to produce rhythmic activity

We first examined pathways downstream of the endogenous clock signal, such as metabolic rhythms and core body temperature. As shown in *Figure 3A–D*, control mice exhibited robust bimodal circadian rhythms of oxygen consumption (VO2), carbon dioxide production (VCO2), heat, and body temperature, while NS-ZB20KO mice displayed decreased peaks of VO2, VCO2, heat and body temperature during the early evening phase and increased peaks of VO2, VCO2, heat and body temperature during ZT22-ZT24 (*Figure 3A–D*). The peaks of these bimodal rhythms were somewhat less pronounced than those of activity rhythms (*Figure 1D–F*), and they appeared to correspond only to the changes in activity patterns. Importantly, these rhythms were still maintained in NS-ZB20KO mice

Next, we crossed the *Nestin-Cre*; Zbtb20fl/+ mice to the Zbtb20$^{fl/fl}$; *Per2$^{Luc}$* knock-in reporter mice (*Yoo et al., 2004*) and monitored *Per2*-luciferase oscillation in SCN tissues and in lung and liver explants. As expected, the circadian period, amplitude and phase of PER2:LUC expression from these tissues were comparable in WT and NS-ZB20KO mice (*Figure 3—figure supplement 1*). These data suggested that the impairment of the circadian oscillator might be milder than the loss of rhythm in behavioral activity.

## ZBTB20 is required for the expression of the known clock output gene *Prokr2*

To investigate the molecular effects of the loss of *Zbtb20* on the disruption of SCN coupling or output, we analyzed the expression of various well-known, abundantly expressed SCN genes, including endogenous core circadian genes and genes involved in the intercellular coupling of the SCN region (*Aton et al., 2005*; *Bedont et al., 2014*; *Cheng et al., 2002*; *Harmar et al., 2002*; *Hatori et al., 2014*; *Kramer et al., 2001*; *Lee et al., 2015*; *Li et al., 2006*; *Maywood et al., 2011*; *Prosser et al., 2007*; *Yamaguchi et al., 2013*). The levels of the circadian core components *Per1*, *Per2*, *Nr1d1*, *Nr1d2*, *Bmal1*, *Clock*, *Rora*, and *Rorb* in the SCN were comparable between control and NS-ZB20KO

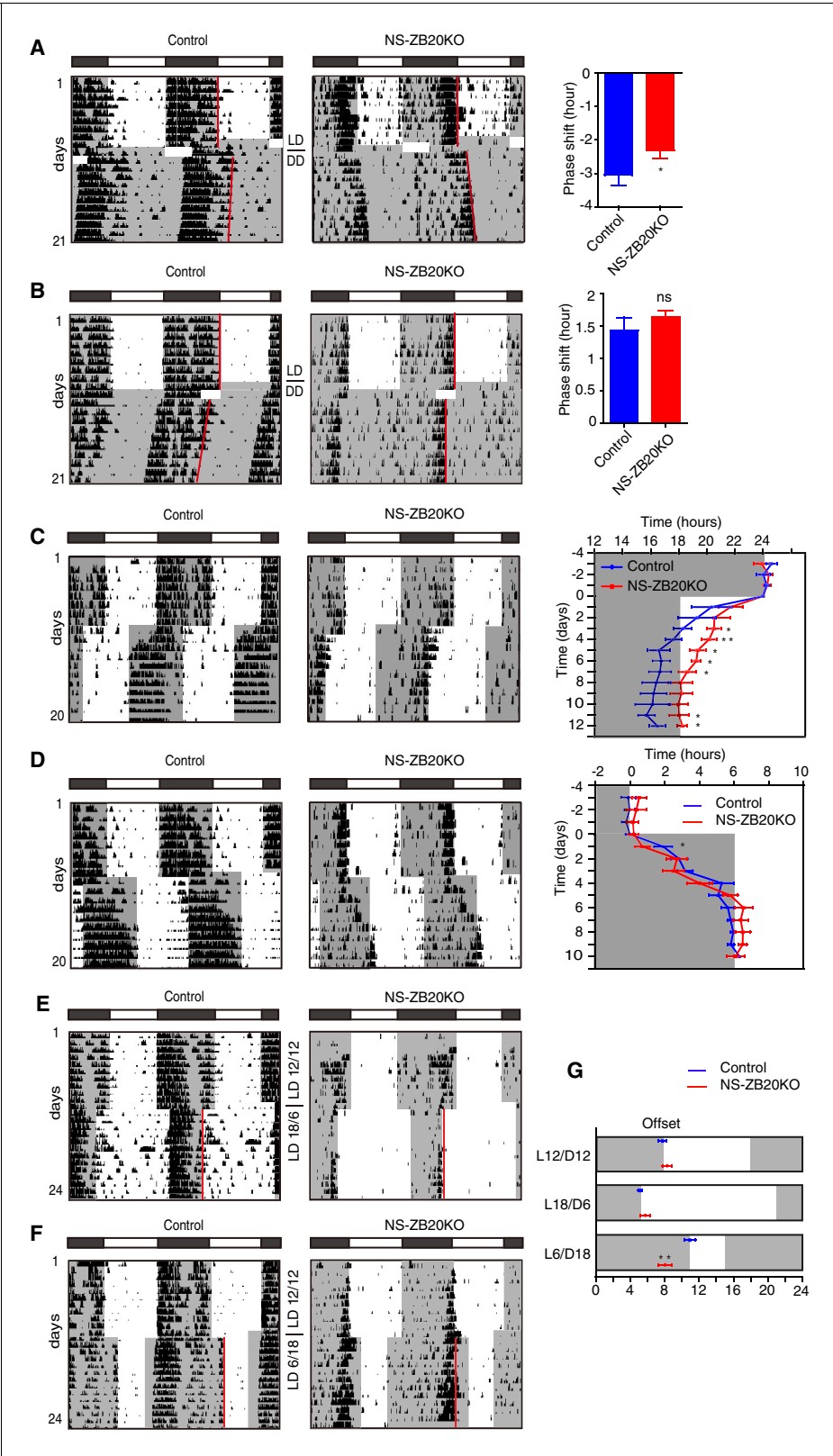

**Figure 2.** NS-ZB20KO mice show defects in light responsiveness. (**A,B**) Representative actograms of wheel-running activity of control (n = 6) and NS-ZB20KO (n = 11) mice subjected to a 6 hr or 4 hr light pulse at CT12 (**A**) or CT20 (**B**), including the mean phase shift in activity. Negative numbers are phase delays, and positive numbers are phase advances. Values are expressed as the mean ± SEM. *p=0.0475 by Student's t-test. (**C,D**) Representative actograms of wheel-running activity of control (n = 6) and NS-ZB20KO (n = 6) mice subjected to a 6 hr phase advance (**C**) and delay (**D**), including

*Figure 2 continued on next page*

*Figure 2 continued*
average activity offset in the 6 hr phase advance (**C**) and delays (**D**). Values are expressed as the mean ± SEM, *p<0.05, **p<0.01 by Student's *t*-test. (**E**–**G**) Representative actograms of wheel-running activity of control (n = 6) and NS-ZB20KO (n = 6) mice transferred to 18L/6D (**E**) and 6L/18D (**F**) from 12L/12D. (**G**) Average phase relations of activity offset (red line) to different photoperiods. Gray areas show the dark phase. Values are expressed as the mean ± SEM (n = 6 for control mice and NS-ZB20KO mice, **p=0.001174 by Student's *t*-test).

mice at CT8 and CT20 (*Figure 4A*). In addition, circadian oscillation of BMAL1 protein in the NS-ZB20KO deficient SCN was normal (*Figure 4—figure supplement 1A*), suggesting that the circadian oscillator was less affected in NS-ZB20KO mice, consistent with the above conclusion. The expression of *Dbp*, a clock output gene, and *Cry2* were elevated only at CT20 (*Figure 4A*). NS-ZB20KO mice showed no obvious effects on the transcript levels of *Vip*, *Avp*, *Prok2* or *Grp*. The expression

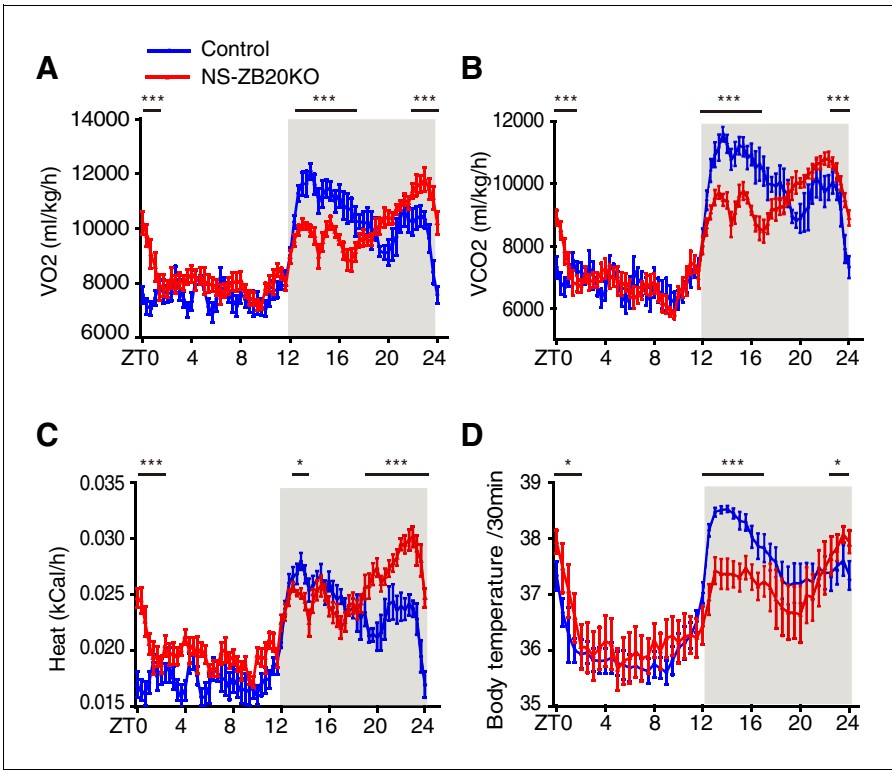

**Figure 3.** Loss of ZBTB20 alters metabolism and body temperature rhythms. (**A**,**B**) Temporal profiles of the $O_2$ consumption rate (VO2) (**A**) and $CO_2$ production rate (VCO2) (**B**) monitored over three consecutive 24 hr cycles for control and NS-ZB20KO mice kept in LD. The result from 3 consecutive days of the LD cycle was accumulated into 20 min bins and averaged to produce one 24 hr profile. (**C**) Temporal profiles of the whole-body heat production calculated by the volumes of $O_2$ and $CO_2$ consumption in the CLAMS chambers. (**A**–**C**) Numbers of mice: for control, n = 12; for NS-ZB20KO, n = 9, *p<0.05, **p<0.01, ***p<0.001 by one-way ANOVA with multiple comparisons and Tukey's test. (**D**) Temporal profiles of core body temperature measured for 3 days with subcutaneously implanted miniature telemetry. Number of mice for controls, n = 9, for NS-ZB20KO, n = 5. The data, as the mean ± SEM, are shown as diurnal averages in 20 or 30 min bins, *p<0.05, **p<0.01, ***p<0.001 by one-way ANOVA with multiple comparisons and Tukey's test (also see *Figure 3—figure supplement 1*).

The following source data and figure supplement are available for figure 3:

**Source data 1.** Data for temporal profiles of the O2 consumption rate, CO2 production rate, heat production, and body temperature in control and NS-ZB20KO mice.

**Figure supplement 1.** Effect of *Zbtb20* depletion by *Nestin-Cre* on the SCN, lung and liver explants.

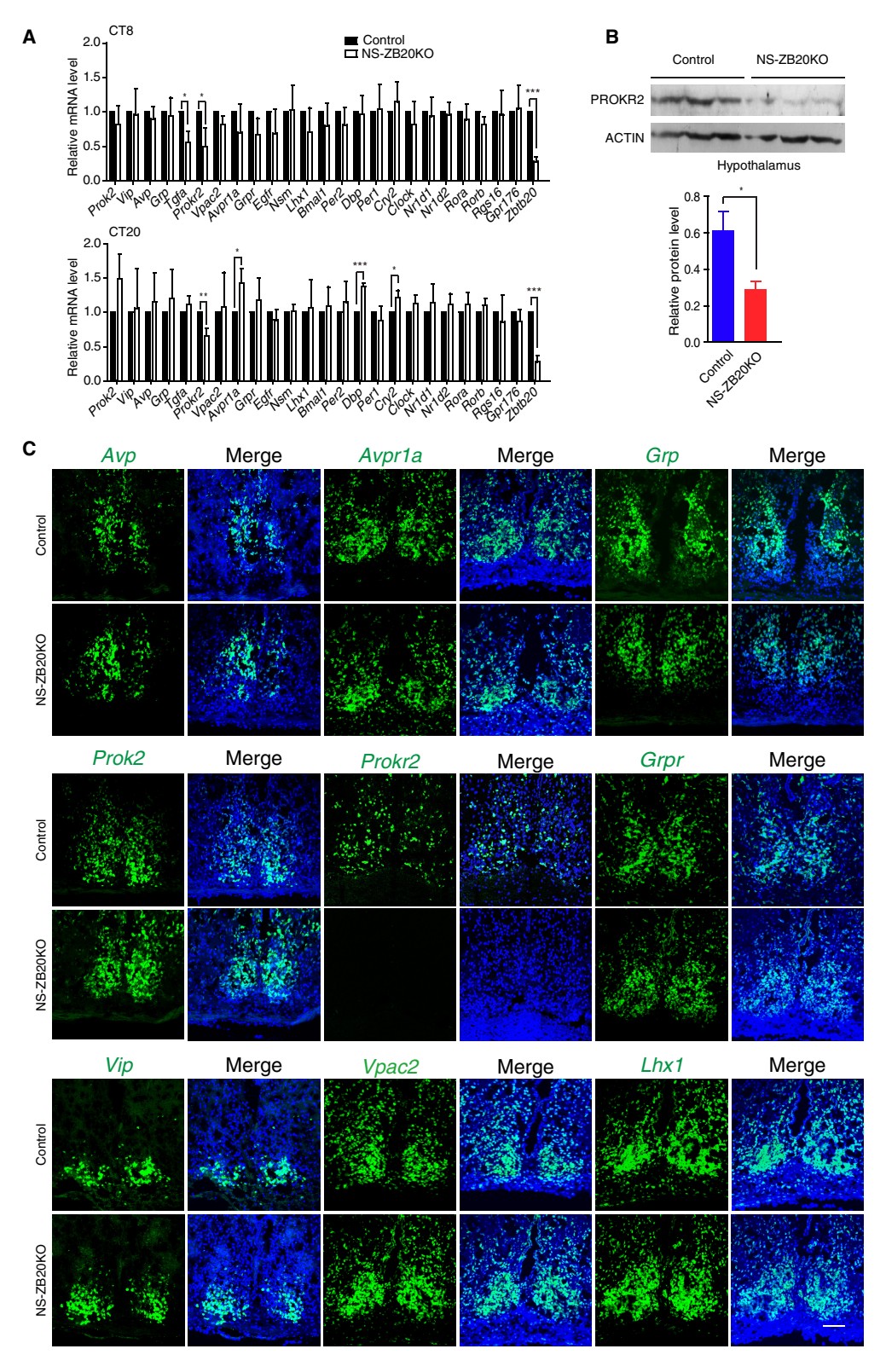

**Figure 4.** *Prokr2* expression decreased in the SCN of NS-ZB20KO mice. (**A**) Average transcript levels of key factors and core circadian clock genes in the SCN from control (n = 4) and NS-ZB20KO mice (n = 4). Values are expressed as the mean ± SEM, *p<0.05, **p<0.01, ***p<0.001 by Student's *t*-test. (**B**) Western blot analysis demonstrated that PROKR2 protein level was significantly lower in the hypothalamus of NS-ZB20KO mice (Control mice n = 5; NS-ZB20KO mice n = 5). Values are expressed as the mean ± SEM, *p<0.05 by Student's *t*-test. (**C**) Fluorescence in situ hybridization in the SCN for *Vip*,

*Figure 4 continued on next page*

Figure 4 continued

*Vpac2, Avp, Avpr1a, Prok2, Prokr2, Grp, Grpr*, and *Lhx1*. Slices from at least three mice per group, Scale bar, 100 μm (also see *Figure 4—figure supplements 1–2*).

The following source data and figure supplements are available for figure 4:

**Source data 1.** Primer list for Q-PCR.

**Source data 2.** Data for qPCR analysis in control and NS-ZB20KO mice SCN.

**Figure supplement 1.** Loss of *Zbtb20* does not alter the expression patterns of BMAL1, AVP and VIP protein in SCN.

**Figure supplement 2.** Normal morphogenesis of the SCN in NS-ZB20KO mice.

levels of *Nsm*, **Lhx1**, *Rgs16* and *Gpr176*, recently reported as key regulators in SCN neurons (*Bedont et al., 2014*; *Doi et al., 2011, 2016*; *Hatori et al., 2014*; *Lee et al., 2015*), remained equivalent between NS-ZB20KO and control mice (*Figures 4A*). The transcript level of Tgfa was only affected in NS-ZB20KO mice at CT8, without obvious differences at CT20. The transcript level of **Avpr1a** was only affected in NS-ZB20KO mice at CT20 (*Figure 4A*). It is unclear whether the changes in these neuropeptides are consequences or the cause of the altered activity. Importantly, the accumulation of mRNA for **Prokr2**, a G protein-coupled receptor for PROK2 (*Prosser et al., 2007*), was dramatically and consistently attenuated in NS-ZB20KO mice at both CT8 and CT20. We next examined the level of PROKR2. Western blot analysis using hypothalamus extracts confirmed that depletion of *Zbtb20* led to reduced PROKR2 protein, consistent with the changes that we observed in *Prokr2* mRNA levels (*Figures 4B*).

To verify the above results and to observe the distribution of these peptides in SCN neurons, we performed in situ hybridization. A deficiency in *Zbtb20* resulted in a remarkable decrease in *Prokr2* mRNA in the SCN, while no significant changes were observed in the mRNA accumulation or distribution of *Vip, Vpac2, Avp, Avpr1a, Prok2,Grp, Grpr* or *Lhx1* (*Figures 4B*), suggesting that the loss of *Zbtb20* in SCN did not severely affect its differentiation. In addition, rhythmic alterations in AVP and VIP content were comparable between control and NS-ZB20KO mice (*Figure 4—figure supplement 1B and C*). Immunofluorescence assays using antibodies targeting BMAL1, AVP and VIP showed intact morphogenesis of the SCN along the rostral-caudal axis in NS-ZB20KO mice (*Figure 4—figure supplement 1D–F*). Moreover, Nissl staining and neuronal nuclear antigen (NeuN) antibody, a marker for mature, post-mitotic neurons, were employed to test the physiological status of the SCN (*Herculano-Houzel and Lent, 2005*; *Mullen et al., 1992*). Histology and NeuN staining showed no obvious differences between control and NS-ZB20KO mice (*Figure 4—figure supplement 2A and B*). Furthermore, we found no obvious difference in the SCN region in terminal deoxynucleotidyl transferase dUTP nick-end labeling (TUNEL)-positive cells or in glutamic acid decarboxylase (GAD) 65/67, markers for GABAergic neurons (*Figure 4—figure supplement 2C and D*). Altogether, these results strongly suggested that the morphology and development of the SCN region remained intact, that ZBTB20 has strong effects on the positive regulation of *Prokr2* expression, and that the clock oscillator was not several impaired.

## The regulation of *Prokr2 by* ZBTB20 is required for the early evening activity

Because the loss of *Zbtb20* resulted in a phenotype reminiscent of mice without PROK2 or PROKR2 signaling (*Li et al., 2006*; *Prosser et al., 2007*), we reasoned that ZBTB20 might alter locomotor rhythms by affecting either *Prok2, Prokr2* expression or PROK2, PROKR2 signaling. Considering that ZBTB20 is a transcription factor and can bind to core elements in target genes to regulate gene transcription (*Xie et al., 2008*; *Zhang et al., 2012*), we first analyzed the genomic sequence for the *Prokr2* gene, as previously reported (*Nielsen et al., 2014*). The sequences 5'-ATTTTTA-3' and 5'-GATACAG-3' called the M1 and M2 sites, were found in the *Prokr2* promoter (*Figure 5A*). We used chromatin immunoprecipitation (ChIP) to evaluate the potential ZBTB20 binding sites of *Prokr2*. Chromatin from NS-ZB20KO hypothalami was used as a control for background binding, and

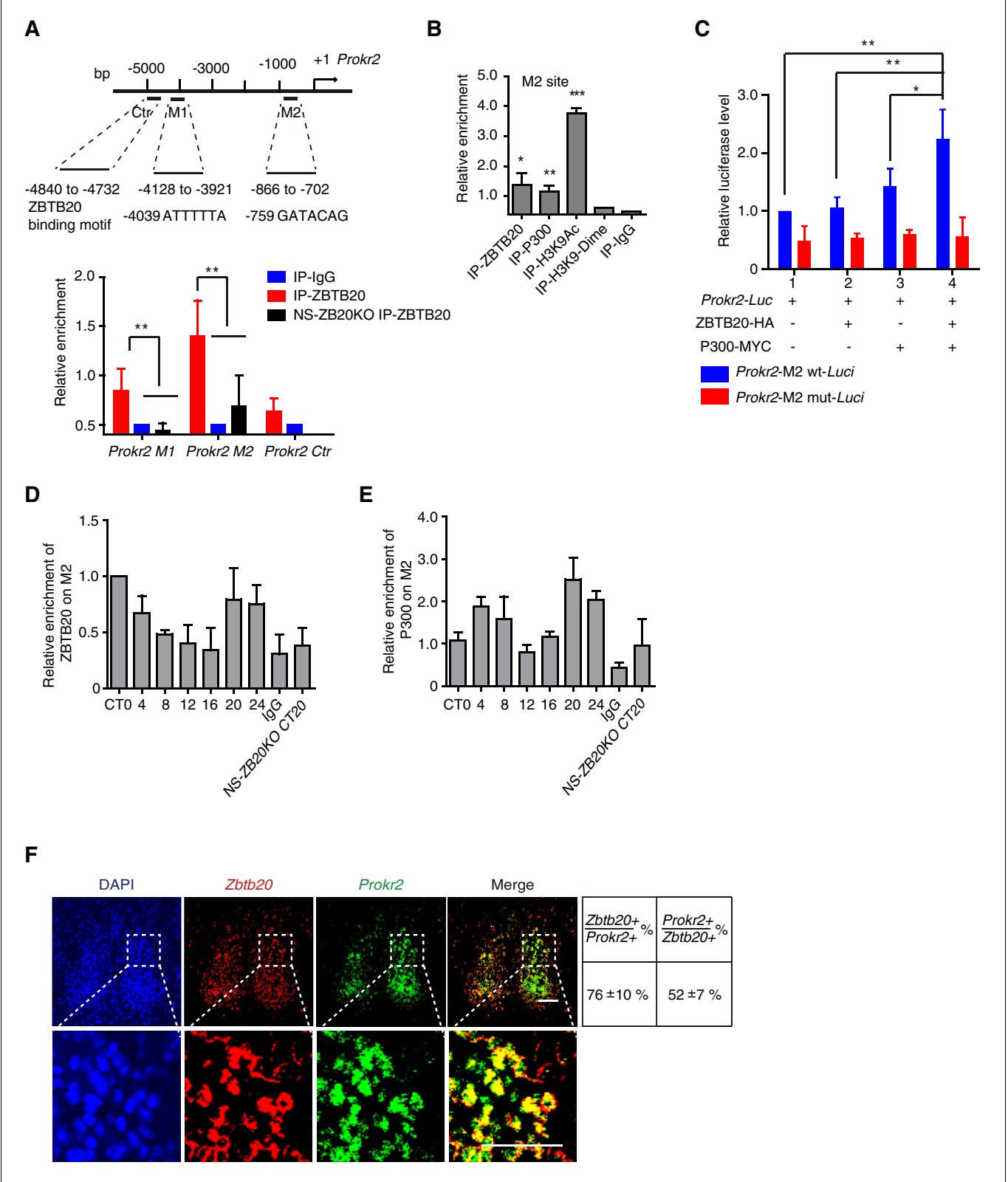

**Figure 5.** ZBTB20 up-regulates *Prokr2* gene transcription. (A) Bioinformatics analysis showed two potential ZBTB20 binding sites in the *Prokr2* promoter region (top), and a ChIP assays showed the enrichment of ZBTB20 on the *Prokr2* promoter in hypothalamus (bottom). (B) ChIP analysis for the enrichment of ZBTB20, P300, H3K9Ac, and H3K9 dimers on the M2 site in the hypothalamus. (C) Overexpression of ZBTB20 and P300 enhanced *Prokr2* promoter activity in SCN 2.2 cells. (D,E) Daily binding levels of ZBTB20 (D) and P300 (E) on the M2 site were determined using ChIP at CT0, CT4, CT8,

*Figure 5 continued on next page*

*Figure 5 continued*

CT12, CT16, CT20, and CT24. (A,B,D) Each hypothalamus sample for the CHIP assay was collected from 5 mice, and the experiments were repeated three independent times. (C) The experiment was repeated four independent times. (A–C) Values are expressed as the mean ± SD, *p<0.05, **p<0.01, ***p<0.001 by Student's *t*-test. (F) Representative double fluorescence in situ hybridization for *Zbtb20* and *Prokr2* in mouse SCN. Scale bar, 100 μm, (also see *Figure 5—figure supplement 1*).

The following source data and figure supplement are available for figure 5:

**Source data 1.** Primer list for CHIP.
**Source data 2.** Data for CHIP analysis.
**Figure supplement 1.** Non- or low-rhythm expression of *Zbtb20*.

ZBTB20 target sites were immunoprecipitated from control hypothalamus using antibodies against ZBTB20, with anti-immunoglobulin G as a negative control. As shown in *Figure 5A*, relatively high enrichment of ZBTB20 binding occurred in the M2 region, which is 700 bp upstream of the transcriptional start site (TSS), while another site, M1, located at 4.7 kb upstream from the TSS, showed a slight but significant signal (*Figures 5A*).

ZBTB20 has been identified as a transcriptional repressor (*Xie et al., 2008*; *Zhang et al., 2012*); however, most of the putative ZBTB20 binding sites overlapped with P300-bound regions in the hippocampus (*Nielsen et al., 2014*), suggesting an active signature. We thus scanned the enrichment region for P300 binding, from the TSS to -5 kb from *Prokr2,* using an anti-P300 antibody, and we found that P300 and ZBTB20 co-occupied overlapping M2 sites (*Figure 5B*). In addition, acetylation of histone H3K9 also occurred at the M2 site (*Figure 5B*), suggesting the coordinated regulation of active histone markers. To identify the transcription factor that mediated the regulation of the *Prokr2* gene by ZBTB20, we examined the ability of ZBTB20 to synergize with P300 in the regulation of *Prokr2* transcription in the cell line SCN 2.2 (*Earnest et al., 1999*). ZBTB20 enhanced P300-mediated transcriptional activity on a *Prokr2* promoter reporter (*Figure 5C*). The synergistic effects of ZBTB20 and P300 were abolished when the M2 site was mutated (*Figure 5C*). Interestingly, the enrichment profiles of ZBTB20 binding to the M2 site showed diurnal rhythms (*Figure 5D*) together with P300 occupancy (*Figure 5E*), although the accumulation of *Zbtb20* mRNA and ZBTB20 from the hypothalamus revealed very weak diurnal rhythms (*Figure 5—figure supplement 1*). In addition, the expression of *Zbtb20* and *Prokr2* overlapped in the SCN neurons (*Figure 5F*). These results, combined with the reduced *Prokr2* expression in NS-ZB20KO SCN neurons, strongly suggested that ZBTB20 regulated *Prokr2* expression and that *Prokr2* might be a downstream target of ZBTB20.

To determine whether ZBTB20 acts with *Prokr2* to regulate the evening activity, we used a rescue strategy. We generated a *Cre*-inducible adeno-associated virus (AAV) vector carrying double-floxed *Prokr2-mCherry* or *mCherry* only. Upon transduction, *Cre*-expressing cells inverted the *Prokr2-mCherry* ORF or *mCherry* in an irreversible manner and thereby activated sustained *Prokr2-mCherry* or *mCherry* under the strong, constitutively active elongation factor 1A (EF-1A) promoter (*Figure 6A and B*), which only occurred in CRE-positive cells, resulting in the restricted expression of PROKR2 or mCHERRY in *Zbtb20*-knockout cells. As an additional control, we also injected control mice with AAV carrying double-floxed *Prokr2-mCherry* ORF or *mCherry*. The virus was stereotaxically injected into the bilateral SCN of 6- to 8-week-old mice. Thus, all of the mice, whether control or NS-ZB20KO mice, shown in *Figure 6C* had received injections of AAV-double-floxed *Prokr2-mCherry* ORF or *mCherry*, and the expression was confirmed by the mCHERRY signal (*Figure 6D*).

To assess the effects on circadian locomotor activity, mice were individually housed in wheel-running cages in the LD cycle for at least 10 days. To exclude inter-cage variations (the friction force of each cage is different), mice were kept in the same wheel-running cage after injection. To avoid variation in the strength of activity of each mouse, each mouse served as its own control by comparing the measurements before and after injection. Unfortunately, most of the injected NS-ZB20KO mice died because NS-ZB20KO mice were unhealthy and surgical procedures could also lead quite easily to death. In the surviving mice, we assessed the activity distribution from day 3 to day 10 before injection and from day 10 to day 30 after injection. We found that the counts of activity distribution

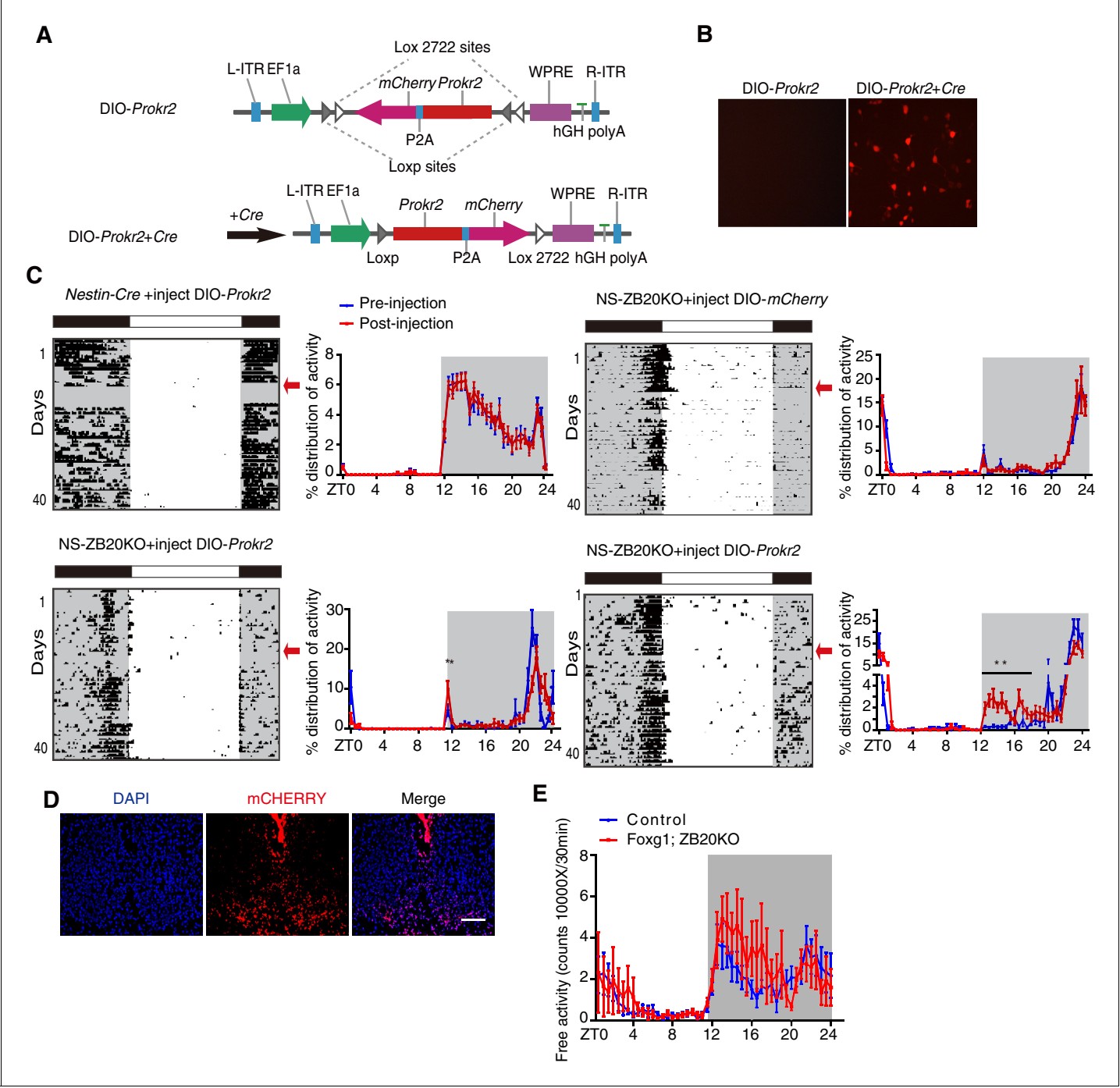

**Figure 6.** ZBTB20 maintains evening behavioral rhythms partly via PROKR2. (**A**) *DIO-Prokr2. Prokr2-mCherry* ORF was flanked by double loxp. (**B**) CRE-mediated expression of *Prokr2-mCherry* in CRE-positive cells. (**C**) Representative actograms of wheel-running activities from controls that received *DIO-Prokr2*, and NS-ZB20KO that received *DIO-mCherry* and *DIO-Prokr2*. Each mouse was kept in same wheel running cage, and its locomotor activity was recorded before and after recovery from surgery. Arrows indicated the time of AAV injection. Percentage distributions of wheel-running activity were quantified from locomotor assay, before injection (blue, wheel running activity from day 3 to day 10 before injection) and after injection (red, wheel running activity from day 10 to 30 after injection). The results are plotted as the mean ± SEM, **p<0.01 by by one-way ANOVA with multiple comparisons and a Tukey's test, (n = 10 for control mice received *DIO-Prokr2;* n = 10 for NS-ZB20KO received *DIO-mCherry*; n = 10 for NS-ZB20KO received *DIO-Prokr2*, most of NS-ZB20KO mice were died, and three mice were partly rescued. (**D**) mCHERRY expression of NS-ZB20KO mice with *DIO-Prokr2* virus injection. Scale bar, 100 μm. (**E**) Temporal profiles of spontaneous activity recorded by CLAMS for control and Foxg1; ZB20KO. The spontaneous activity from 3 consecutive day of the LD cycle was accumulated to 30 min bins and averaged to produce one 24 hr profile. The number of mice for CLAMS: control n = 7; *Foxg1; ZB20KO* n = 7. Each point represents the mean ± SEM, (also see *Figure 6—figure supplement 1*).

*Figure 6 continued on next page*

*Figure 6 continued*

The following source data and figure supplement are available for figure 6:

**Source data 1.** Data for temporal profiles of wheel-running activity.

**Source data 2.** Data for temporal profiles of spontaneous activity in control and Foxg1; ZB20KO mice.

**Figure supplement 1.** Expression of ZBTB20 in SCN from control and Foxg1; ZB20KO mice.

during the evening phase significantly increased only in NS-ZB20KO mice injected with *Prokr2-mCherry,* although the evening activity was only partly rescued (*Figure 6C*, two bottom columns).

*Zbtb20* is expressed in hippocampal neurons, pyramidal neurons, striatum neurons, and dorsolateral pallium neurons outside the SCN (*Mitchelmore et al., 2002*; *Rosenthal et al., 2012*), and thus, it is possible that *Zbtb20* deletion in *Nestin* loci other than the SCN was responsible for the behavioral phenotype of NS-ZB20KO mice. We generated mice lacking *Zbtb20* in other head regions by crossing *Foxg1-Cre* (*Hébert and McConnell, 2000*), which is expressed by E7.5 in the facial ectoderm; head ectoderm; otic, optic, nasal and oral epithelia; forebrain; and foregut, with floxed *Zbtb20* alleles (*Zbtb20*fl/fl), termed as Foxg1; ZB20KO mice. There was no obvious difference in *Zbtb20* expression in the SCN between Foxg1; ZB20KO and control mice (*Figure 6—figure supplement 1*). We measured the circadian rhythms of spontaneous locomotor activity in Foxg1; ZB20KO mice using CLAMS. The behavior of these mice showed obvious bimodal activity, although the total activity counts were higher than those of control mice (*Figure 6E*), suggesting that loss of *Zbtb20* in *Foxg1* loci was not responsible for the unimodal activity of NS-ZB20KO mice. These results, combined with previous reports (*Li et al., 2006*; *Prosser et al., 2007*), suggested that PROKR2 signaling was directly correlated with the early evening activity. ZBTB20 is therefore a critical regulator of *Prokr2* in the SCN for early evening activity.

## Genome-wide identification of ZBTB20 target genes

The loss of *Zbtb20* led to a severe neurobehavioral phenotype, but the over-expression of PROKR2 caused only partial rescue of the early evening activity in NS-ZB20KO mice. Moreover, because ZBTB20 is a transcription factor, we attempted to assess the extent of the involvement of ZBTB20 in the regulation of gene expression in the SCN region at both the transcript level and the protein level. First, we performed RNA-Seq on the total RNA isolated from the SCN of control and NS-ZB20KO mice at CT9 and CT11 to explore the origins of the evening activity peak. As shown in *Figure 7A*, of the 20,349 genes expressed in the SCN, NS-ZB20KO mice showed significant changes in 205 genes at CT9 and 427 at CT11 (p value<0.05) (*Figure 7—source data 1*, *Figure 7—source data 2*). GO_BP enrichment and KEGG analysis were performed using the DAVID bioinformatics resource (http://david.abcc.ncifcrf.gov/). The differentially expressed genes in the most over-represented functional categories according to DAVID (GO_BP) included mitochondrion, cytoplasmic vesicle, and translation (*Figure 7A*, *Figure 7—source data 1*, *Figure 7—source data 2*)

Two significant KEGG pathways were identified: the neurodegeneration pathway including Parkinson's disease (PD), Alzheimer's disease (AD) and Huntington's disease (HD), and the oxidative phosphorylation pathway. The genes *Ndufb4, Ndufa2, Uqcr10, Ndufb7, Cox6a2,* and *Atp5g3* were considered critical, showing a high degree (Kappa statistics, 1.0) of sharing by these pathways. These pathways are involved in neurodegenerative disorders, indicating that progressive disorders might begin with mitochondrial dysfunction in NS-ZB20KO mice. In keeping with the evening activity peak at CT12, nocturnal animals should store energy to anticipate activity at CT12. However, we found that the most significantly enriched pathways, including protein catabolic processes and mitochondria, were disrupted in NS-ZB20KO mice at CT11, immediately before CT12 (*Figure 7A*). We speculated that the enrichment of these pathways might be interpreted as a cause for the loss of the early evening activity because there was a lack of energy storage or mitochondrial dysfunction in NS-ZB20KO mice. Then, we compared the genes showing dynamic alteration between CT9 to CT11 between the NS-ZB20KO mice and the control mice (*Figure 7B*). Seven genes, including the dopamine receptor activity-related gene Drd1a, were down-regulated between CT9 to CT11 in the

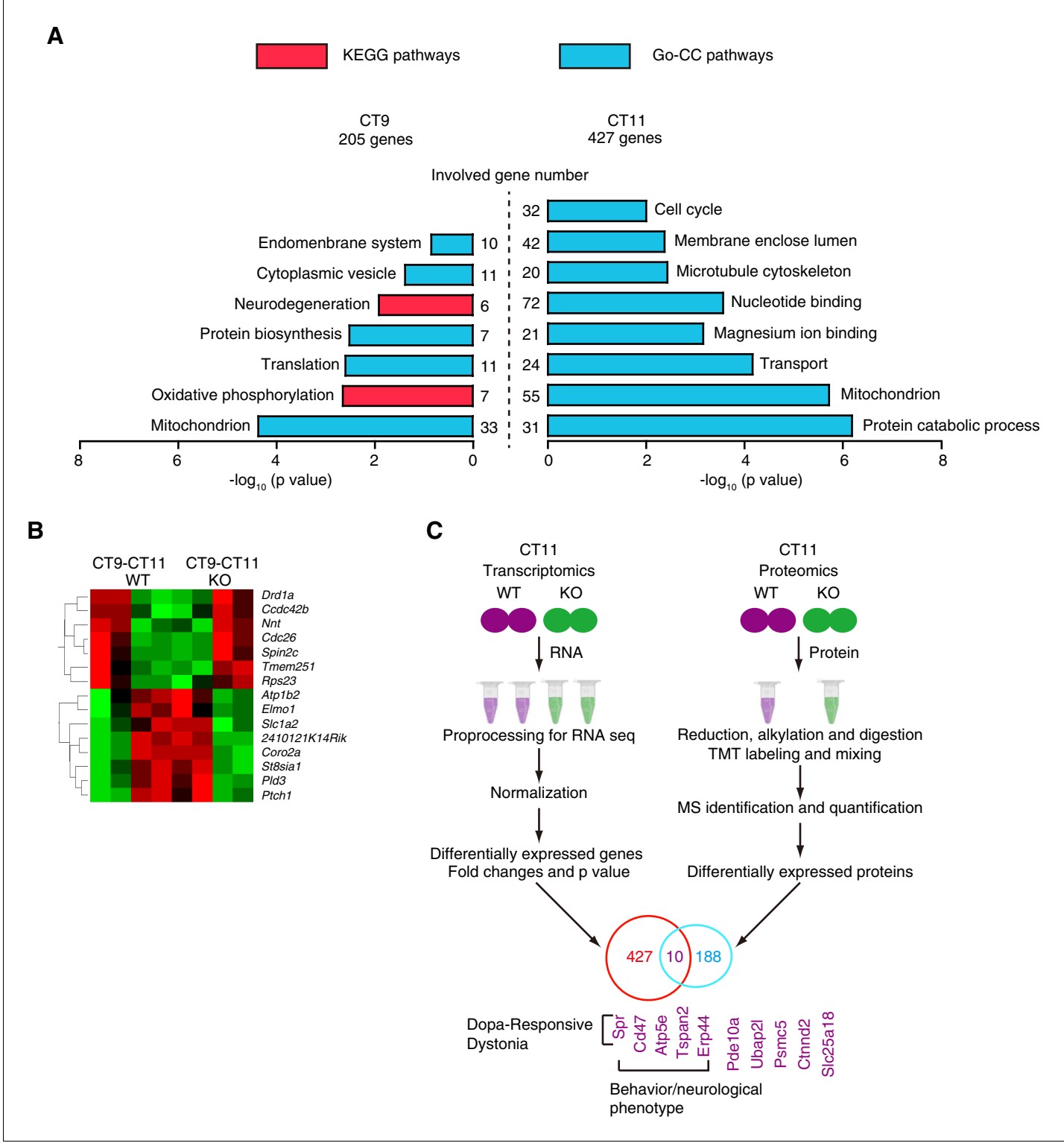

**Figure 7.** Genome-wide identification of ZBTB20 target genes. (**A**) Diagrams depicting the functional category of differentially regulated transcripts between different genotypes at CT9 (left) and CT11 (right) from control and NS-ZB20KO mice. (**B**) Hierarchical clustering analysis of transcripts (q value <0.05) showed opposite trends between control (WT) and NS-ZB20KO (KO) mice from time point CT9 to CT11. High levels are shown in red, and low levels are shown in green. (**A,B**) Each SCN sample for RNA-seq was collected from 3 mice; and each group was repeated two times. (**C**) Overlapping altered transcripts in NS-ZB20KO mice between the RNA-seq data and the quantitative proteomics data. Each SCN sample for quantitative proteomics was collected from 5 mice, (also see *Figure 7—source data 1–4*).

*Figure 7 continued on next page*

*Figure 7 continued*

The following source data is available for figure 7:

**Source data 1.** Expressed genes were significantly changed in NS-ZB20KO mice at CT9.
**Source data 2.** Expressed genes were significantly changed in NS-ZB20KO mice at CT11.
**Source data 3.** proteins were upregulated in the NS-ZB20KO mice.
**Source data 4.** proteins were downregulated in the NS-ZB20KO mice.

control mice, while these genes presented the opposite trend in the NS-ZB20KO mice. Conversely, the ATPase activity-related gene Atb1b2, the vesicle-mediated transport gene Elmo1, the glutamate clearance gene Slc1a2, and cell motility-related genes were obviously down-regulated from CT9 to CT11 in NS-ZB20KO mice, as shown on the heat maps (*Figure 7B*).

Quantitative proteomics using the tandem mass tag (TMT) labeling technique can identify proteins differentially expressed under different experimental conditions (*Thompson et al., 2003*). We used duplex TMT labeling reagents (TMT-126 and TMT-127) to label proteins in the SCN tissues from both control and NS-ZB20KO mice. Proteins with TMT labeling ratios 20% higher or 20% lower than 1.0 were considered to be upregulated or downregulated, respectively. In total, 89 proteins were identified as upregulated and 99 as downregulated in the NS-ZB20KO mice (*Figure 7—source data 3*, *Figure 7—source data 4*), and these genes were enriched for involvement in intracellular signaling cascades, cellular macromolecular complex assembly, mitochondrion, and neurotransmitter transport. Finally, we integrated the NS-ZB20KO RNA-Seq data sets with proteomic data sets to identify genes altered at both the transcription and translation levels. We found that 10 genes overlapped: Spr, Cd47, Atp5e, Tspan2, Erp44, Pde10a, Ubap2l, Psmc5, Ctnnd2 and Slc25a18 (*Figure 7C*). Previous studies found that mutations in SPR (sepiapterin reductase) result in DOPA-responsive dystonia (*Blau et al., 2001*; *Steinberger et al., 2004*). ATP5E is associated with several diseases, including isolated ATP synthase deficiency and mitochondrial complex III deficiency, nuclear type 5. Collectively, these data suggest that the loss of *Zbtb20* disrupted multiple pathways for metabolism, neurodegeneration, development and the cell cycle.

## Discussion

Locomotor assays are extensively used to study the underpinnings of the circadian clock in various organisms. Compared to the number of identified core clock genes, very few circadian output genes have been isolated, indicating that clock output factors might be important for development such that mutations are lethal, or that these genes compensate for each other, i.e., are functionally redundant. In this study, we demonstrated that NS-ZB20KO mice showed a time-dependent loss of early evening activity and substantially increased morning activity. Our results strongly suggest that severe reduction in the *Prokr2* expression level resulted in the absence of early evening activity in NS-ZB20KO mice. More importantly, the phenotype in NS-ZB20KO mice can be partly rescued by the overexpression of PROKR2 in the SCN region, and the deletion of *Zbtb20* using *Foxg1-Cre* in other head regions beyond the thalamus and hypothalamus did not phenocopy the phenotypes of NS-ZB20KO mice, indicating that ZBTB20-mediated *Prokr2* signaling is critical for rhythmic behavioral outputs from the SCN.

It is well known that neuropeptides in pacemakers are critical for rhythmic behavioral output. For instance, the loss of vasoactive intestinal polypeptide (VIP) dramatically impaired circadian rhythms and behavior (*Aton et al., 2005*), while mice lacking AVP receptor V1a showed significantly elongated activity time (*Li et al., 2009*). Recent studies have demonstrated that V1a and V1b signals confer intrinsic resistance to external perturbations on the SCN (*Yamaguchi et al., 2013*). However, some neuropeptides can compensate for each other, such as GRP and AVP (*Maywood et al., 2011*). Furthermore, the *Prokr2* gene encodes a G protein-coupled receptor for PROK2 that has also been implicated in linking circadian pacemakers to behavioral output without affecting the molecular oscillator (*Prosser et al., 2007*). PROK2 is a peptide expressed in subpopulations of SCN neurons,

including those producing AVP, VIP, and GRP, and PROK2 expression is induced by light pulses that reset the clock (*Cheng et al., 2002*; *Masumoto et al., 2006*). This may account for our observation of impaired entrainment in NS-ZB20KO mice (*Figure 2*).

The loss of *Prok2* or *Prokr2* has a phenotype surprisingly similar to that observed in NS-ZB20KO mice. Unlike the phenotypes observed in the TGFa signal (*Kramer et al., 2001*) and the *Clc* signal (*Kraves and Weitz, 2006*), the loss of *Zbtb20* resulted in the profound disruption of the activity/rest cycle under both LD and DD conditions. Because the circadian period was only subtly affected (*Figure 1C*) and the expression levels of core clock genes in the SCN were comparable between the control and NS-ZB20KO mice (*Figure 4A*), we hypothesized that ZBTB20 mainly targets the output pathway. In addition, metabolic and body temperature rhythms were maintained with a possible compromise in activity (*Figure 3A–D*). Furthermore, most circadian genes in peripheral tissues showed unperturbed rhythms (data not shown). This also fits with an output role for *Zbtb20*. A relative redistribution of activity from early to late circadian night was suggested in *Prokr2* null mice (*Prosser et al., 2007*). We thus asked whether a hyperactivity peak during early morning was a 180° reversal from the early evening peak in NS-ZB20KO mice. We found that light pulses during CT12-CT18 induced a phase delay despite entrained behavior being weak in NS-ZB20KO mice (*Figure 2A*). This suggested that CT12–CT18 was subjective early evening for NS-ZB20KO mice. Moreover, light pulses during CT20-CT24 caused a phase advance, suggesting that CT20-CT24 was late night for NS-ZB20KO mice (*Figure 2B*). We therefore suggested that the loss of *Zbtb20* resulted in the ablation of evening behavioral rhythms rather than a shift from the early evening peak to an early morning peak, although the mechanism underlying the burst of early morning activity remains unclear.

There is reliable evidence showing that ZBTB20 regulates the direct expression of *Prokr2*. In addition to showing that the phenotype of NS-ZB20KO mice resembles that observed in *Prokr2* null mice (*Prosser et al., 2007*), we found that ZBTB20 bound to the *Prokr2* promoter and activated *Prokr2* transcription in SCN 2.2 cells along with P300. In the absence of *Zbtb20*, we observed a strongly decreased level of *Prokr2* expression. Moreover, in our rescue experiments with overexpression of *Prokr2* in NS-ZB20KO mice, abnormal behavior was partially restored. Thus, it is clear that ZBTB20 mediated PROKR2 signaling is essential for the bimodal activity output. Similar models have been proposed in previous reports in which morning and evening oscillators were organized in the SCN and regulated morning behavior and evening behavior (*Inagaki et al., 2007*; *Kon et al., 2014*). We also generated AVP and VIP *Cre* mice in order to delete *Zbtb20* in these neurons. However, mice lacking *Zbtb20* in AVP or VIP neurons showed no obvious behavioral defects (data not shown). Because the NS-ZB20KO mice are generally sick, it is difficult to determine how much of the circadian defect is specifically due to the requirement for ZBTB20-*Prokr2* function and how much is simply due to fact that NS-ZB20KO mice suffer from a myriad of neurological and metabolic maladies. However, we generated mice with the deletion of *Zbtb20* in neurons expressing *Foxg1*, a forebrain marker, and we found that Foxg1; ZB20KO mice had normal circadian activity. Because Zbtb20 is enriched in the SCN and because Prokr2 and Prok2 are also expressed in multiple types of neurons, coupling among these neurons might play an important role in the early evening activity output. Together with data from previous studies on Prok2 and Prokr2 (*Cheng et al., 2002*; *Li et al., 2006*; *Prosser et al., 2007*), we propose that ZBTB20 is a convincing candidate as a circadian oscillator to produce coherent rhythmic outputs.

Perturbation of the communication between pacemakers and downstream targets can disrupt the orchestrated rhythmic behavior output and lead to disorders, including neurodegenerative disorders (*Videnovic et al., 2014*) and sleep disorders (*Sehgal and Mignot, 2011*), and they are also involved in driving the disease process itself. Although AD, PD, and HD have distinct phenotype-genotype signatures, each also exhibits overlapping symptoms that could, at least partly, be based on circadian dysfunction. Interestingly, the progressive deterioration of behavioral rhythms, as well as the alteration in the core body temperature of NS-ZB20KO mice, mimics the clinical indicators of AD, PD and HD (*Videnovic et al., 2014*).

In summary, our work demonstrated that ZBTB20 is a critical transcription factor for maintaining early evening activity, as opposed to inhibiting the strength of early morning activity. Because ZBTB20 plays a very important role in the induction of *Prokr2* expression in the SCN, which contributes to behavioral rhythms, it will be interesting to use various specific *Cre* transgenic mice to determine which subgroup of SCN neurons is essential for early evening activity.

# Materials and methods

## Animals

NS-ZB20KO and Foxg1; ZB20KO mice were generated by crossing the loxP-flanked exon 6 of *Zbtb20* allele mice (*Xie et al., 2008*) with the nerve-system *Nestin-Cre* (Jackson stock #003771, RRID:IMSR_JAX:003771) mice (*Chang and Guarente, 2013*) or *Foxg1-Cre* (*Hébert and McConnell, 2000*), respectively. *Nestin-Cre; Zbtb20*$^{fl/fl}$; *Per2*$^{luc}$ mice were generated by crossing *Nestin-Cre; Zbtb20*$^{fl/+}$ mice to the *Zbtb20*$^{fl/fl}$; *Per2*$^{Luc}$ knock-in reporter mice (RRID IMSR_JAX:006852) (*Yoo et al., 2004*). They were backcrossed for at least six generations with C57BL/6J (Jackson stock #000664, RRID:IMSR_JAX:000664) mice. All of the mice were housed in specific pathogen-free animal facilities with *ad libitum* access to food and water under a LD cycle (lights on at 08:00, lights off at 20:00). All of the animal procedures were approved by the Animal Care and Use Committee of the Model Animal Research Center, Nanjing University and Animal Ethics Committee of Second Military Medical University.

## Locomotor activity analysis

For wheel-running activity assay, as previously described (*Xu et al., 2007*), six- to eight-week-old mice were individually housed in cages equipped with running wheels, and they were initially entrained to a LD cycle for at least 7 days, followed by constant darkness for several weeks. Wheel rotation was recorded using ClockLab software (Actimetrics, RRID:SCR_014309).

For the light pulse-induced shift experiments, as previously reported (*Thresher et al., 1998*), the mice were kept in an LD cycle for 9 days and were placed in DD and exposed to a 6 hr or 4 hr light pulse at either CT12 or CT20. Phase shifts in behavioral activity were quantified as time differences between regression lines of activity offsets before and after the light application.

For the 'jet lag' experiments, as previously reported (*Mieda et al., 2015*; *Yamaguchi et al., 2013*), the mice were entrained to a LD cycle for 9 days. On the 10th day, the LD cycle was either advanced for 6 hr or delayed for 6 hr, and the animal behavior was recorded for 11 days following the LD cycle shift. The average numbers of days to re-entrain to the shifted LD cycle were quantified as the offset in the time of wheel-running activity.

For the different photoperiod experiments, a detailed protocol was previously reported (*Inagaki et al., 2007*). The mice were firstly kept in a standard LD cycle for 9 days (lights on at 08:00, lights off at 20:00) and then were released into the 18L/6D cycle, which was altered by advancing the light on and delaying the light off by 3 hr (lights on at 05:00, lights off at 23:00) or were released into the 6L/18D cycle, which was changed by delaying the light on by 3 hr and advancing the light off (lights on at 11:00, lights off at 17:00). The offset of nocturnal activity were analyzed using the activity rhythms of the previous 7 days in different photoperiods.

The free spontaneous activity was measured with the Comprehensive Lab Animal Monitoring System (CLAMS; Columbus Instruments) and by miniature telemetry (TA-F10, Data Sciences International, New Brighton, Minnesota), as previously described (*Li et al., 2006*; *Wang et al., 2010*; *Xu et al., 2007*).

Clocklab software was used to analyze the time of wheel-running activity onset, the time of wheel-running activity offset, the distribution of wheel-running activity, the total wheel-running activity, and a chi-square periodogram (*Harmar et al., 2002*; *Kon et al., 2014*). The percentage of each 30 min activity over 24 hr of total activities was calculated and plotted with respect to circadian time (CT) or Zeitgeber time (ZT) (*Li et al., 2006*).

## Luminescence recording and data analysis

*Per2*$^{Luc}$ knock-in reporter mice were mated with NS-ZB20KO mice. These mice were kept in a standard LD cycle. Explants were prepared and cultured as described (*Liu et al., 2014*; *Yoo et al., 2004*). One hour before lights off, explants were briefly prepared and immediately placed in Hank's balanced salt solution. Then, explants were then cultured with 1.2 ml DMEM (Product No.D2902, Sigma), supplemented with 2% B27 (Product No. 17504–044, Gibco), 10 mM HEPES (pH 7.2), antibiotics (100 U/ml penicillin, 100 U/ml streptomycin, 0.1 mM luciferin (Promega), 4.5 g/l glucose, and 4.2 mM NaHCO3. SCN and liver were cultured on the Millicell culture membranes (0.4 µM, 30 mm diameter, Millipore). Bioluminescence was mounted over 10 min intervals with the LumiCycle

(LumiCycle, Actimetrics) (*Yamazaki et al., 2000*). Date was analyzed using the LumiCycle Analysis software as described (*Liu et al., 2014*; *Wang et al., 2010*).

## Metabolic rhythm measurement and analysis

VO2 and VCO2 were measured with a Comprehensive Lab Animal Monitoring System (CLAMS), as previously described (*Liu et al., 2014*). The mice were housed in individual cages within a temperature-controlled room (21 ± 1°C). Following an adaptation period of 3 days, the mice were continuously recorded for another 3 days in 20 min time bins. The rate of energy expenditure (heat) was calculated using the formula according to manufacturer's protocol: heat = (3.815 + 1.232*VCO2/VO2)*VO2.

## Core body temperature monitoring

Core body temperature was measured using implantable telemetry (TA-F10, Data Sciences International). The mice were kept in normal LD cycles for 1 week, and the telemeter was inserted into the abdominal cavity by sterile surgery. The ambient temperature was 21°C. Following a week of convalescence, the core body temperature was continuously recorded for 4 days by a receiver board (Data Sciences International) beneath the cage. The data were plotted as daily average temperature at 30 min intervals for at least 8 mice of each genotype (*Gerhart-Hines et al., 2013*).

## Fluorescence in situ hybridization

The mice were euthanized by $CO_2$ asphyxiation and were immediately perfused with saline, followed by 4% paraformaldehyde. Brains were dissected and post-fixed in 4% paraformaldehyde for several hours, following overnight incubation in 30% sucrose. Fluorescence in situ hybridization assay was performed, according the manufacturer's procedures, with the following specific cRNA probes, as described by the Allen Institute for Brain Science (http://www.brain-map.org/): for *Prok2*, nucleotides 10–537 (NCBI Reference Sequence: NM_001037539.2); for *Prokr2*, nucleotides 1469–2533 (NM_144944.3), as described in (*Cheng et al., 2002*); for *Vip*, nucleotides 190–717 (NM_011702.3); for *Vpac2*, nucleotides 1506–2386 (NM_009511.2); for *Avp*, nucleotides 1–492 (NM_009732.2); for *Avpr1a*, nucleotides 1781–2070 (NM_016847.2); for *Grp*, nucleotides 88–854 (NM_175012.4); for *Grpr*, nucleotides 1314–2079 (NM_008177.3); for *Lhx1*, nucleotides 1567–2623 (NM_008498.2); and for *Zbtb20*, nucleotides 2057–2730 (NM_019778.2). Digoxigenin-labeled or fluorescence-labeled antisense RNA probes were transcribed with T7 or SP6 RNA polymerases, respectively (Roche). Fluorescence staining was obtained using tyramide signal amplification plus a fluorescence system (TSA, PerkinElmer), according to the manufacturer's protocol. Double fluorescent in situ hybridization utilized standard digoxigenin-labeled *Zbtb20* and fluorescein-labeled *Prokr2* cRNA probes, along with TSA. The cell nuclei were visualized by DAPI staining (Sigma).

## TUNEL assay

Brain slides were processed with ApopTag Fluorescein In Situ Apoptosis Detection Kit (Millipore) following the manufacturer's instructions.

## SCN collection, RNA extraction, RT-PCR, and mRNA expression analyses

The detailed methods for SCN tissue collection were previously described (*Savelyev et al., 2011*). SCN tissue was pooled from three animals per condition. RNA isolation and RT-PCR were performed as previously described (*Wang et al., 2010*). The quality and quantity of RNA were examined using a Nanodrop 1000 spectrophotometer (Thermo Fisher Scientific). The relative levels of each transcript were determined by Q-PCR after RT-PCR, and they were normalized to the corresponding *Gapdh*. The primer sequences are provided in *Figure 4—source data 1*.

## Generation and stereotaxic injection of AAV

Coding regions of *Prokr2* were amplified from C57BL/6J mouse cDNA by PCR and were cloned into the AAV-2 ITR-containing plasmid pAAV-EF1a-double floxed-mCherry-WPRE-HGHpA (provided by Obio Technology). The vector was confirmed by sequencing. Recombinant AAV containing double *floxed-Prokr2-mCherry* ORF or *mCherry* was packaged using a triple-transfection, helper-free

method and was purified by Obio Technology. The titers of recombinant AAV supernatant were determined by Q-PCR: *DIO-Prokr2*: $2.1 \times 10^{12}$; and *DIO-mCherry*: $2.3 \times 10^{12}$ genome copies/ml.

Bilateral stereotaxic injections of AAVs were performed with a glass micropipette and an air pressure injection system, as previously described (*Mieda et al., 2015*). Virus solution at a volume of 1.0 µl was injected at each SCN site (bregma -0.48 mm, lateral ± 0.2 mm, ventral 5.1 mm). After 10 min of rest, the needles were removed. To exclude the variations in each wheel-running cage (the friction force of each cage is different), mice were kept in wheel-running cages, and wheel-running activity was recorded for at least 10 days during LD cycles before injection. Several days after surgery, each mouse was housed in the same wheel-running cage and was continuously recorded for several weeks. Each mouse served as its own control, and the percentage distribution of wheel-running activity was analyzed before injection and after injection.

### RNA sequencing and analysis

SCN tissues were collected from control or NS-ZB20KO mice at CT9 and CT11, respectively. Each SCN tissue was pooled from three animals per condition, and each group was repeated for two times. RNA was sequenced in the WuXi Genome Center (Shanghai, China). Reads were mapped by TopHat (RRID:SCR_013035). Expression values were normalized by RPKM. A gene was supposed to be unexpressed if it's value, from our RNA-Seq, was less than 3. Genes with values less than 3 were removed in the following analysis. The figure was produced by heatmap.2 package in R.

### Quantitative proteomics

Proteins from control and NS-ZB20KO CT11 mouse SCN pools (5 mice for each genotype) were extracted with RIPA buffer in the presence of fresh protease inhibitor cocktail (Roche) and were quantified with BCA protein assay (Thermo Fisher). The same amounts of proteins from two samples were individually precipitated using cold acetone and were solubilized with 8 M urea in 50 mM HEPES (pH = 8.0). The proteins (100 µg each) were reduced with dithiothreitol, alkylated with iodoacetamide, and digested with endoproteinase Lys-C (Wako, Japan) for 3 hr at room temperature and again with sequencing grade modified trypsin (Promega) after diluting to 2 M urea with 50 mM HEPES (pH = 8.0). The resulting peptides were labeled with TMT-126 and TMT-127 labeling reagents, respectively (Thermo Fisher), and were mixed at a 1:1 ratio. The samples were fractionated in a C18 reverse-phase column at a high pH. The fractionated samples were dried and cleaned with C18 ZipTip and were analyzed in an Orbitrap Elite mass spectrometer (Abrahamson and Moore) with high-energy collision dissociation (HCD). MS/MS spectra were searched with Proteome Discoverer software, version 1.3 (Thermo Fisher), against a UniProt mouse database. The false discovery rate (FDR) was set to 1% to obtain highly confident identification. The TMT reporter ions (126/127) from MS/MS spectra were used for relative quantification. Reverse labeling was also performed to increase the coverage of and confidence in protein quantification.

### Immunofluorescence, western blot, and chromatin immunoprecipitation

The mice were euthanized, and the brain tissue was harvested as indicated by the conditions. Immunofluorescence staining, western blotting and chromatin immunoprecipitation were performed as previously described (*Lee et al., 2015*; *Shi et al., 2013*; *Xu et al., 2007*). The antibodies used were the following: mouse anti-ZBTB20 9A10 (1:1,000, home made), mouse anti-BMAL1 (1:1,000, home made), goat anti-PROKR2 (Santa Cruz Biotechnology, 1:400, Cat# sc-54317 RRID:AB_2253158), rabbit anti-AVP (Millipore,1:4000, Cat# AB1565 RRID:AB_11212336); rabbit anti-VIP (Immunostar, 1:2000, Cat# 20077 RRID:AB_572270); rabbit anti-P300 (Santa Cruz Biotechnology, 1:400, Cat# sc-584 RRID:AB_2293429); mouse anti-NeuN (Millipore, 1:100, Cat# MAB377 RRID:AB_2298772); goat anti-GAD65/67 (Santa Cruz Biotechnology, 1:400, Cat# sc-7513 RRID:AB_2107745); rabbit anti-acetyl-histone H3 (Lys9) (Millipore, 1:500, Cat# 07–352 RRID:AB_310544); and rabbit anti-dimethyl-histone H3 (Lys9) (Upstate Biotechnology, 1:500, Cat# 07–441 RRID:AB_310619). The NIH Image J (RRID:SCR_003070) software was used for the quantification of the band from Western blots. The primer sequences for CHIP are provided in *Figure 5—source data 1*.

## Cell culture, transfection, and luciferase report assay

SCN2.2 cells (RRID:CVCL_D050) were cultured in MEM containing 10% FBS, 0.4% glucose, and 2 mM L-glutamine in 96-well dishes (*Earnest et al., 1999*). Transfection assay was performed by using Lipofectamine 2000 (Thermo Fisher Scientific) according to the manufacturer's instructions. Luciferase reporter gene assays were carried out with a Dual-Report assay system (Promega); 50 ng *Prokr2-Luc*, 50 ng ZBTB20, 50 ng P300, 1 ng Renilla pRL-TKvector were added and the pCMV-Tag2B plasmid was brought to the same amount.

## Statistical analysis

In *Figure 5A–D*, *Figure 1—figure supplement 1A*, *Figure 4—figure supplement 2C*, and *Figure 5—figure supplement 1A,D*, results are shown as mean ± SD; and other results are presented as mean ± SEM. No statistical analysis was used to determine appropriate sample size. All statistical tests were carried out with GraphPad Prism software (GraphPad, RRID:SCR_002798). Each experiment was repeated at least three independent times or was performed with several mice each group. Due to the size of SCN is small; each SCN tissue was pooled from at least three animals per condition. In *Figure 1D–F*, *Figure 3*, and *Figure 6C*, differences between genotypes were determined using one-way ANOVA with multiple comparisons and a Tukey's post-test. For other results, paired or unpaired student's *t*-test was performed to compare the difference between two groups with or without homogeneity of variance, respectively. Values were considered significantly different with $p < 0.05$.

## Acknowledgements

We appreciate Yong Zhang (University of Nevada) for reading the paper and giving insightful comments. We thank members of the Xu and Zhang's laboratory for their assistance. This work was supported by grants from the National Science Foundation of China (31230049, 31630091 to YX; 81130084, 31671219 to WZ), Royal Society-Newton Advanced Fellowship (NA150373 to YX), and a project funded by the priority Academic Development of Jiangsu Higher Education Institutions.

## Additional information

### Competing interests

JST: Reviewing editor, *eLife*. The other authors declare that no competing interests exist.

### Funding

| Funder | Grant reference number | Author |
| --- | --- | --- |
| National Science Foundation of China | 31230049 | Ying Xu |
| National Science Foundation of China | 81130084 | Weiping J Zhang |
| Royal Society Newton Advanced Fellowship | NA150373 | Ying Xu |
| National Science Foundation of China | 31630091 | Ying Xu |
| National Science Foundation of China | 31671219 | Weiping J Zhang |

The funders had no role in study design, data collection and interpretation, or the decision to submit the work for publication.

### Author contributions

ZQ, YX, Conception and design, Acquisition of data, Analysis and interpretation of data, Drafting or revising the article, Contributed unpublished essential data or reagents; HZ, HL, Generated mouse models, Provided intellectual input, Contributed unpublished essential data or reagents; MH, Performed bioinformatics assay, Acquisition of data, Analysis and interpretation of data; GS, ZL, PX,

Contributed to the mouse work, Assisted with data analysis, Contributed unpublished essential data or reagents; WW, Performed bioinformatics assay, Analysis and interpretation of data; GX, YZ, Performed mass assay, Acquisition of data, Contributed unpublished essential data or reagents; LY, GH, JST, Provided intellectual input, Analysis and interpretation of data; WJZ, Analysis and interpretation of data, Contributed unpublished essential data or reagents

### Author ORCIDs
Pancheng Xie, http://orcid.org/0000-0001-6311-6092
Joseph S Takahashi, http://orcid.org/0000-0003-0384-8878
Weiping J Zhang, http://orcid.org/0000-0003-4727-2380
Ying Xu, http://orcid.org/0000-0002-6689-7768

### Ethics
Animal experimentation: This study was performed in strict accordance with the recommendations in the Guide for the Care and Use of Laboratory Animals by Association for Assessment and Accreditation of Laboratory Animal Care International. All of the animals were handled according to approved institutional animal care and use committee protocols (# YX2, # 2009-0088) of the Model Animal Research Center, Nanjing University and Animal Ethics Committee of Second Military Medical University. The protocols were approved by the IACUC committee of the Model Animal Research Center, Nanjing University and Animal Ethics Committee of Second Military Medical University. All surgery was performed under IACUC approved anesthesia. Every effort was made to minimize suffering.

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
