## [Decision Letter]

Thank you for submitting your article "Loss of ZBTB20 impairs circadian output and leads to unimodal behavioral rhythms" for consideration by *eLife*. Your article has been favorably evaluated by a Senior Editor and three reviewers, one of whom is a member of our Board of Reviewing Editors. The following individuals involved in review of your submission have agreed to reveal their identity: Louis Ptáček (Reviewing Editor and Reviewer #1), Hitosbi Okamura (Reviewer #2), and Orie Shafer (Reviewer #3).

The reviewers have discussed the reviews with one another and the Reviewing Editor has drafted this decision to help you prepare a revised submission.

Summary:

In this interesting manuscript Qu and colleagues investigate the effects of the loss of the zinc finger transcription factor ZBT20 on circadian activity rhythms in mice and make a case that this transcription factor acts as an output factor in the central clock. Understanding how the central circadian pacemaker links molecular timekeeping with neuronal outputs is a central question in chronobiology and of intense interest to the field. Using a Nestin-Cre mediated knockdown of ZBT20 throughout the nervous system (ZBT20KO), the authors present compelling evidence that the loss of this transcription factor results in locomotor rhythm abnormalities that are highly reminiscent of the abnormalities seen previously in mice lacking the neuropeptide Prokineticin-2 (Prok2) or it's receptor (Prokr2). The authors provide strong evidence of a link between ZBT20 and Prok2 signaling by showing that Prokr2 levels are dramatically lowered in the SCN ZBT20KO mice and that ZBT20 acts a positive transcriptional regulator of Prokr2 expression in cultured suprachiasmatic nucleus (SCN) cells. The evidence supporting this link is very strong. The evidence that these observations provide specific insights into circadian output pathways in the SCN is less strong (see below).

Essential revisions:

1) A central concern is the fact that the ZBT20KO mice appear to be sick and grossly abnormal with respect to brain development and metabolism. As the authors concede in the manuscript, ZBT20KO mice display myriad abnormalities in addition to the circadian activity phenotypes described. For example, ZBT20KO mice have significantly reduced brain size and weight and a significant proportion of the ZBT20KO die before reaching four months of age. Thus, despite the very nice link the authors have made between ZBT20 and Prokr2, it is difficult to know how much of the circadian defects described are due specifically to the requirement for ZBT20/Prokr2 function in the central clock, and how much was simply due to fact that the ZBT20KO were suffering from myriad neurological and metabolic maladies. This might be circumvented by generating other lines using more-specific drivers but this would require many months of additional work which is deemed outside the scope of the current manuscript.

The one attempt to address the SCN specific function of ZBT20/Prokr2, the over-expression of Prokr2 in the ZBT20KO SCN, was complicated by the fact that the majority of the mice did not survive the viral injection surgery, further evidence that these mice are quite sick. The authors mention attempting ZBT20 knockdowns in cell types that are more specific to circadian timekeeping (VIP and AVP expressing cells), but report that these mice had normal circadian activity rhythms. These negative results could be consistent with an alternative explanation for the central results: the circadian defects displayed by the ZBT20KO mice are simply the result of the widespread neurological and metabolic defects caused by knockdown of ZBT20 in Nestin-expressing neurons throughout the nervous system. The link between ZBTB20 and Prok2 signaling would be interesting in itself but the potential link to a circadian output makes the manuscript more interesting. However, caveats re: the fact that the mice are sick which could contribute to altered activity need to be discussed.

2) It is already reported that ZBTB20 knock out causes a genesis of specific type of cells in the pituitary (Cao et al., Nat Commun 7:11121, 2015), and plays a crucial role for the morphogenesis of hippocampus (Rosenthal et al., Hippocampus 22:2144-56, 2012). Since authors use Nes-CRE, which is expressed neuronal/glial progenitor cells, it is highly possible that ZBTB20 is deleted in developing CNS. Thus the effect of ZBTB20/Nes-CRE on the morphogenesis of the SCN should carefully be checked. One serial Nissl stained sections is shown in a Figure 4—figure supplement 2, yet this does not justify whether morphogenesis is unperturbed. Moreover, immunohistochemical analyses (Figure 4—figure supplement 2) are all suffering from poor experimental quality. For example, the coronal level of GAD-immunohistochemistry should be harmonized in WT and NZ-ZB20KO. The histology data should be convincingly reproduced at more high resolution images. Thus the authors should produce a whole immunohistochemical series of SCN from the entire rostral-caudal axis, by using antisera to AVP and VIP proteins or by other convincing markers.

3) Although Figure 4 shows average transcript levels of key regulators in the SCN, this does not include Rgs16 or Gpr176, both of which are known to work on SCN network as the other genes that the authors tested. The authors are therefore recommended to add Rgs16 and Gpr176 to Figure 4 RT-PCR analysis.

4) The in situ hybridization trials of authors to show the known expressed genes are also encouraged, since this provides a method for examining the affected genes morphologically after the deletion of ZBTB20. Unfortunately, presented quality of in situ analyses (Figure 4 and Figure 4—figure supplement 1 and Figure 4—figure supplement 2; Figure 5 and Figure 5—figure supplement 1; Figure 6 are all poor experimental quality. The authors are therefore recommended to use conventional in situ hybridization assay with RI or DIG. Since the behavioural data use adult mice, the authors should analyze adult mice for checking whether the SCN is intact.

---

## [Author Response]

*1) A central concern is the fact that the ZBT20KO mice appear to be sick and grossly abnormal with respect to brain development and metabolism. As the authors concede in the manuscript, ZBT20KO mice display myriad abnormalities in addition to the circadian activity phenotypes described. For example, ZBT20KO mice have significantly reduced brain size and weight and a significant proportion of the ZBT20KO die before reaching four months of age. Thus, despite the very nice link the authors have made between ZBT20 and Prokr2, it is difficult to know how much of the circadian defects described are due specifically to the requirement for ZBT20/Prokr2 function in the central clock, and how much was simply due to fact that the ZBT20KO were suffering from myriad neurological and metabolic maladies. This might be circumvented by generating other lines using more-specific drivers but this would require many months of additional work which is deemed outside the scope of the current manuscript.*

*The one attempt to address the SCN specific function of ZBT20/Prokr2, the over-expression of Prokr2 in the ZBT20KO SCN, was complicated by the fact that the majority of the mice did not survive the viral injection surgery, further evidence that these mice are quite sick. The authors mention attempting ZBT20 knockdowns in cell types that are more specific to circadian timekeeping (VIP and AVP expressing cells), but report that these mice had normal circadian activity rhythms. These negative results could be consistent with an alternative explanation for the central results: the circadian defects displayed by the ZBT20KO mice are simply the result of the widespread neurological and metabolic defects caused by knockdown of ZBT20 in Nestin-expressing neurons throughout the nervous system. The link between ZBTB20 and Prok2 signaling would be interesting in itself but the potential link to a circadian output makes the manuscript more interesting. However, caveats re: the fact that the mice are sick which could contribute to altered activity need to be discussed.*

We are grateful to the reviewers for their constructive comments and suggestions. We do acknowledge that it is difficult to know what proportion of the circadian defects was due specifically to the requirement for ZBTB20/Prokr2 function and what proportion was simply due to the fact that NS-ZB20KO mice suffered from myriad neurological and metabolic ailments. It is not currently possible to speculate about the roles of other brain regions, although we generated mice lacking ZBTB20 in neurons expressing Foxg1, a forebrain marker, and these mice showed normal circadian activity. To support this view, we have expanded the Discussion by highlighting this alternative explanation (fourth paragraph).

*2) It is already reported that ZBTB20 knock out causes a genesis of specific type of cells in the pituitary (Cao et al., Nat Commun 7:11121, 2015), and plays a crucial role for the morphogenesis of hippocampus (Rosenthal et al., Hippocampus 22:2144-56, 2012). Since authors use Nes-CRE, which is expressed neuronal/glial progenitor cells, it is highly possible that ZBTB20 is deleted in developing CNS. Thus the effect of ZBTB20/Nes-CRE on the morphogenesis of the SCN should carefully be checked. One serial Nissl stained sections is shown in a Figure 4—figure supplement 2, yet this does not justify whether morphogenesis is unperturbed. Moreover, immunohistochemical analyses (Figure 4—figure supplement 1 and Figure 4—figure supplement 2) are all suffering from poor experimental quality. For example, the coronal level of GAD-immunohistochemistry should be harmonized in WT and NZ-ZB20KO. The histology data should be convincingly reproduced at more high resolution images. Thus the authors should produce a whole immunohistochemical series of SCN from the entire rostral-caudal axis, by using antisera to AVP and VIP proteins or by other convincing markers.*

Several concerns were related to development, regarding whether the finding of abnormal circadian activity reflected perturbed morphogenesis. As suggested, we performed immunofluorescence assays to evaluate the SCN morphogenesis and expression patterns of SCN genes over the entire rostral-caudal axis of the SCN in *NS-ZB20KO* mice, by using antibodies targeting the proteins AVP, VIP and BMAL1. These data are in the new version of Figure 4—figure supplement 1. Moreover, we also performed immunohistochemistry assays to evaluate the morphogenesis of the SCN over the entire rostral-caudal axis of the SCN in *NS-ZB20KO* mice, by using antibodies targeting a mature neuron marker (NeuN) and GAD65/67. These data are shown in the new version of Figure 4—figure supplement 2. The results suggest that the morphogenesis and maturation of the SCN are normal in *NS-ZB20KO* mice. A statement summarizing these morphological observations has been added to the paper (subsection “The regulation of Prokr2 by ZBTB20 is required for the early evening activity”, first paragraph).

*3) Although Figure 4 shows average transcript levels of key regulators in the SCN, this does not include Rgs16 or Gpr176, both of which are known to work on SCN network as the other genes that the authors tested. The authors are therefore recommended to add Rgs16 and Gpr176 to Figure 4 RT-PCR analysis.*

We thank the reviewers for this suggestion. We scanned for binding sites of ZBTB20 on *Rgs16* and *Gpr176* and found no candidate sites. We estimated the expression of *Rgs16* and *Gpr176* at CT8 and CT20 and found that the levels of *Rgs16* and *Gpr176* were less affected after the deletion of ZBTB20. This conclusion has been included in revised Figure 4.

*4) The in situ hybridization trials of authors to show the known expressed genes are also encouraged, since this provides a method for examining the affected genes morphologically after the deletion of ZBTB20. Unfortunately, presented quality of in situ analyses (Figure 4 and Figure 4—figure supplement 1 and Figure 4—figure supplement 2; Figure 5 and Figure 5—figure supplement 1; Figure 6 are all poor experimental quality. The authors are therefore recommended to use conventional in situ hybridization assay with RI or DIG. Since the behavioural data use adult mice, the authors should analyze adult mice for checking whether the SCN is intact.*

I apologize for this issue. A number of technical issues marred the presented quality of the images.

A) Our histology data are produced at high resolution. When these images were buried in a Word document and were then saved as PDF files, the resolution and color of our histology data were markedly distorted. We did not notice this problem when we submitted our previous version of the manuscript. I apologize.

B) As the reviewers noted, the experimental quality of the in situ hybridization is poor. We have replaced these with new images in Figure 4 for *Avp, Avpr1a, Prok2*, and *Grpr* expression.

We performed in situ hybridization assays with DIG-labeled specific cRNA probes, as recommended by the reviewers. Then, we used tyramide signal amplification plus a fluorescence system (TSA, PerkinElmer) to obtain the fluorescence staining, according to the manufacturer’s protocol. For genes with low expression levels, such as *Prokr2*, we chose the TSA system to amplify the signal as described in the Methods section.

The reviewers correctly noted that we used adult mice for checking whether the SCN is intact in relation to the observed behavioral defects. For TUNEL assays, we analyzed both pups at P3, the time point of obvious SCN neuron cell death (Joseph L. Bedont, et al., Cell Reports) (Figure 4—figure supplement 2), and adult mice. We did not show TUNEL staining from adult mice because there are no TUNEL signals in either WT or KO mice.